# Accuracy Boosters: Epoch-Driven Mixed-Mantissa Block Floating Point for DNN Training

Simla Burcu Harma*, Ayan Chakraborty*, Babak Falsafi*, Martin Jaggi*, Yunho Oh†,

*EcoCloud, EPFL *simla.harma@epfl.ch, ayan.chakraborty@epfl.ch babak.falsafi@epfl.ch martin.jaggi@epfl.ch*
†*ComSys*, Korea University *yunho_oh@korea.ac.kr*

*Abstract*—The unprecedented growth in DNN model complexity, size, and amount of training data has led to a commensurate increase in demand for computing and a search for minimal encoding. Recent research advocates Hybrid Block Floating Point (HBFP) to minimize silicon provisioning in accelerators by converting the majority of arithmetic operations in training to 8-bit fixed point. In this paper, we perform a full-scale exploration of the HBFP design space using mathematical tools to study the interplay among various parameters and identify opportunities for even smaller encodings across layers and epochs. Based on our findings, we propose *Accuracy Boosters*, an epoch-driven mixed-mantissa HBFP technique that uses 6-bit mantissas only in the last epoch and first/last layers, and 4-bit mantissas for $99.7\%$ of all other arithmetic operations in training. Using analytic models, we show Accuracy Boosters enable increasing arithmetic density for an HBFP training accelerator by up to $21.3\times$ compared to FP32 and up to $4.4\times$ compared to another SOTA format BFloat16, while preserving or outperforming FP32 accuracy.

## I. INTRODUCTION

Over the past decade, improvements in Deep Neural Network (DNN) algorithms have led to unprecedented growth in model complexity and dataset size and, consequently, the required computational resources to train DNN models. One of the largest DNN models (GPT-3) [2] has 175 billion parameters and requires $3.14 \times 10^{23}$ FLOPs to train. With the slowdown in Moore's law, researchers and vendors have begun to search for alternate ways to improve the arithmetic density of the underlying hardware platforms. Narrower bit-width (with lower precision) number formats [24], [25], [31], [32], [35] have emerged as a promising approach to increase arithmetic density, as well as, reduce the required operand storage and communication bandwidth while maintaining high training accuracy.

Recently there have been several proposals for block floating point [7], [20], [38], a numerical encoding that groups a block of mantissas which rely on only fixed-point arithmetic with a single exponent. Block floating point asymptotically approaches the arithmetic density of fixed point with larger block sizes and naturally lends itself well to mixed-precision hardware where a block with the same number of exponent bits can have a fixed-point datapath which is bitsliced for various multiples of mantissa bit encodings (e.g., the same way as today's CPU cores implement SIMD). While block floating point has been promising in use for inference (e.g., Microsoft Floating Point [6]), most proposals to train with block floating point have either failed to reach its full potential by requiring small blocks and/or just fall short of reaching FP32 accuracy.

One specific proposal, Hybrid Block Floating Point (HBFP) [10], uses a mixed-precision format where the dominant fraction of training which are the dot products, happen in block floating point (e.g., convolutions, matrix multiplications), and FP32 is used for other less frequent operations requiring larger numerical ranges (e.g., activations, regularizations). HBFP simultaneously offers the high accuracy of floating point and the superior hardware density of fixed point, delivering up to $8.5\times$ higher throughput than FP16 with $2\times$ more compact models [11]. Prior work on HBFP only presented a preliminary design space analysis for power-of-two mantissa bit widths (e.g., 2, 4, 8 bits).

In this paper, we make the observation that the parameter space for HBFP is quite rich, presenting several opportunities for further improving efficiency and density in hardware platforms. First, custom accelerators can support non-power-of-two numerical formats, and minimizing the number of bits improves operand storage and communication linearly and arithmetic logic quadratically. Second, there is an interplay between the block size and the number of mantissa bits, allowing for an overall denser numerical format with smaller blocks while maintaining accuracy. Finally, HBFP allows for mixed-mantissa block floating point encodings. Prior work studies training with various HBFP formats in isolation; however, the design space of mixed-mantissa HBFP is yet to be explored.

We fully explore the parameter space of HBFP and show the boundaries of block floating point by studying the interplay between the block size and the number of mantissa bits. To the best of our knowledge, this is the first paper conducting a full design space exploration for training DNNs with block floating point. We show that HBFP6 (HBFP with 6 bits of mantissa) is the smallest HBFP format achieving competitive accuracies with no sensitivity to block size. Our main contribution is the design of Accuracy Boosters, a DNN training mechanism performing a large fraction of epochs in low precision, i.e. HBFP4. Our method improves epoch-wise mixed-precision training by introducing high precision, i.e. HBFP6, to the training process only at the last epoch. Accuracy Boosters enable increasing arithmetic density by up to $21.3\times$ compared to FP32, and up to $4.4\times$ compared to another SOTA format BFloat16, while preserving or outperforming FP32 accuracy.

## II. HBFP PARAMETER SPACE

HBFP is a mixed-precision DNN training technique that uses block floating point for all dot product operations and

FP32 for the rest of the operations, enabling accurate training with dense fixed-point arithmetic. We observe that HBFP is also suitable for inference for the popular CNN and Transformer models without any accuracy loss, in line with prior work on inference with block floating point [6], showing that HBFP is a versatile technique for both training and inference. Prior work on HBFP shows that the area and energy expenditure of HBFP8 is around an order of magnitude lower than FP32 [11]. Exploring the parameter space of HBFP and pushing its boundaries can increase this ratio dramatically.

HBFP has a rich parameter space, including the number of mantissa bits, block size, and the number of exponent bits. The hardware area and energy expenditure of HBFP accelerators are determined by the number of mantissa bits and the block size because the overhead of the exponent bits is negligible due to blocking[1]. One of the key advantages of HBFP is that we can conservatively find a lower bound for the number of exponent bits that covers all of the design space exploration for block size and the number of mantissa bits. Therefore, we work with 10-bit exponents as in prior work [10] and explore the HBFP design space by varying the mantissa bit width and the block size. Once we fix the number of exponent bits, we can vary other parameters, which enables a reconfigurable microarchitecture and gives rise to mixed-mantissa HBFP.

Smaller mantissa bit widths and larger block sizes are key to improving block-floating-point hardware efficiency due to the increasing fraction of fixed-point operations [6]. There is an interplay between the number of mantissa bits and block sizes, allowing for an overall denser numerical format with smaller blocks while maintaining accuracy. This interplay is the result of how the block floating point conversion works. Block floating point shares a single exponent across a block of values using the exponent of the largest element. Since block floating point format does not apply normalization (It is calculated as $2^{exponent} \times 0.mantissa$ instead of $2^{exponent} \times 1.mantissa$), the precision within a block is highly dependent on the largest element in that block, which decides the exponent value. The interval between two consecutive representable numbers is calculated as in Equation 1.

$$interval = \frac{2^{largest\ exponent}}{2^{\#\ of\ mantissa\ bits}} \qquad (1)$$

As the number of elements sharing the same exponent increases, the likelihood of disparity in the magnitude of elements also increases, leading to a precision loss for the small elements in the block. As the number of mantissa bits decreases, the model's sensitivity to the block size increases with the corresponding increase in the interval leading to a higher quantization error. More mantissa bits make the distribution more resilient to the quantization error and larger block size, as each element can be represented more accurately.

HBFP's power footprint is not only a function of the HBFP parameters but also of outside factors. Mixed-precision training has emerged as a popular technique to increase

the fraction of leaner arithmetic formats within the training process, motivating us to explore the design space of mixed-mantissa HBFP; because HBFP provides the opportunity to fix the exponent width and vary the number of mantissa bits across layers and epochs.

For CNN models, prior work indicates that the first convolution layer and the last fully connected layer have a larger impact on the final model accuracy, and keeping these layers in high precision allows for reducing precision in the rest of the layers [3], [24], [34], [39]. The first layer takes the input images and filters the images with several convolution kernels and returns feature maps. Thus, it is critical for the final model to keep the input information fully accurate and to preserve the data in the initial feature map. Similarly, for Transformers, the first layer is the input embedding layer, where input tokens are mapped to dense word vectors. The last layer of DNN models returns a probability distribution over the possible outcomes for the underlying DNN task. The important roles of the first and last layers in DNN models make it crucial to retain information better for these layers.

In addition to layers, each training epoch has a different effect on the final model's accuracy [13], [14]. [28] and [36] show that DNNs first learn low-frequency components, where the frequency is defined for the coordinates of the input space. [36] also empirically show that for CNN models, the high-frequency components have higher complexities and are learned in the last epochs. In light of these findings, we hypothesize that high-frequency functional components are more sensitive to quantization errors. Thus, higher precision is required for the last stage of DNN training, where the optimization occurs after an appropriate amount of generalization in the network. After reaching a certain loss value in low-precision training, switching the tensors to high precision enable the sensitive fine-tuning performed in the final epochs and help increase the accuracy even more.

## III. MINIMIZING HBFP

Our goal is to minimize HBFP to increase the hardware efficiency of training without losing accuracy. For a block size of $576$, even though HBFP4 incurs a $2.4\times$ improvement in area/power relative to HBFP8, it lacks the precision to reach FP32 accuracy. As prior work on HBFP [10], [11] only investigated power-of-two mantissa bits and focused mostly on the design space of HBFP8, the interplay between the number of mantissa bits and the block size is left unexplored. While power-of-two-bit numbers align naturally with the memory structure and encode matrices in a tightly-packed way, non-power-of-two-bit mantissas can improve the arithmetic density even further, as studied by [6] and can be easily integrated into custom accelerators. We investigate the whole design space of HBFP by varying both parameters and claim that reducing the block size will enable reducing the number of mantissa bits, and thus improve hardware efficiency. In this section, we show how to minimize HBFP step by step, give an explanation of the limitations of HBFP and propose a new mixed-precision schema to minimize HBFP further.

---

[1]Even for the block size of 4, HBFP4 with 5-bit exponent is only $1.1\times$ more area-efficient than HBFP4 with 10-bit exponent

To study the relationship between model accuracy and HBFP parameters, we measure the similarity between block-floating-point and FP32 distributions of various tensors using Wasserstein distance, mathematically defined as in Equation 2.

$$W(P,Q) = \inf_{\gamma \in \Pi(P,Q)} \mathbb{E}_{(x,y) \sim \gamma}[||x - y||] \quad (2)$$

where $\Pi(P,Q)$ is the set of all joint distributions $\gamma(x,y)$ whose marginal distributions are equal to $P$ and $Q$. $\gamma(x,y)$ can be interpreted as the amount of mass that must be transported from $x$ to $y$ to transform $P$ to $Q$ [1]. Unlike KL-Divergence, which is commonly used to compare quantized tensors to their full-precision counterparts [6], [26], Wasserstein distance is symmetric, and thus is mathematically a metric. Moreover, because DNNs often deal with distributions where KL Divergence is not defined (or infinite), we need to add a noise term to the model distribution to be able to use KL Divergence, which causes disturbance in the results.

We observe that the tensor distribution is preserved when the elements are converted to block floating point format with 6 bits of mantissa and wider for reasonably large block sizes[2]. Figure 1 shows Wasserstein distances between FP32 and HBFP6 and HBFP4 with various block sizes for the weight tensors of four different layers of ResNet20 trained on CIFAR10. For all the tensors, HBFP6 has a much smaller distance to FP32, and the distances are fairly close to each other for a given tensor across all block sizes. However, the Wasserstein distance of HBFP4 is more than $3.5\times$ higher than HBFP6 across all block sizes, and the distances dramatically increase with the block size. Indeed, the R-squared (the strength of the relationship between two data sets) values between the model accuracy and various Wasserstein distances are around $0.99$, validating the strength of our metric.

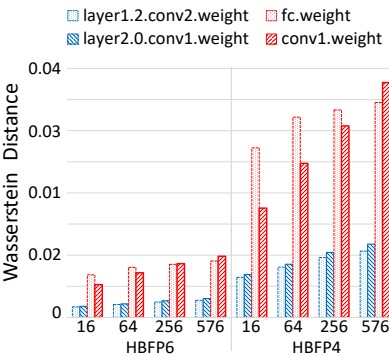

Fig. 1: Wasserstein distance between FP32 and HBFP with various block sizes for various layers.

Even though reducing the block size incurs smaller Wasserstein distances and helps increase the accuracy, HBFP4 still fails to reach FP32 accuracy because it does not have enough precision to minimize the loss and has a high generalization error. [22] introduce a methodology to visualize loss landscapes

[2]Block sizes of up to 256 already achieves more than 95% of the maximum hardware benefit for HBFP6

in order to better understand the effect of loss landscapes on generalization. Figure 2 shows log-scale loss landscapes for various configurations, sliced along the x-axis (y=0) for simplicity. The center of the plot corresponds to the current state of the minimizer, and the axis parameterizes a random direction with filter-wise normalization. HBFP4 converges to a much worse minimum compared to HBFP6 and FP32 indicating poor accuracy. Although the minimum of HBFP4 is flat, it does not indicate better generalization because the minimum itself is much worse.

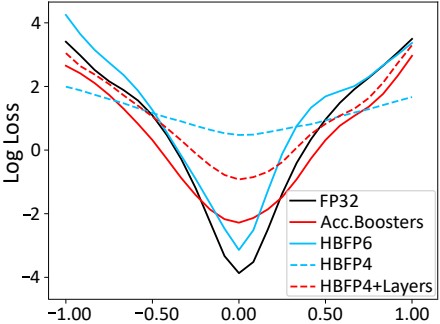

Fig. 2: Loss landscapes of ResNet20 on CIFAR10 for various configurations, sliced along the x-axis.

Following the insights from prior work, we study the effect of the first and last layers of CNNs on model accuracy. The dotted and solid red lines in Figure 1 show the first and last layers, respectively, and it is clear that these layers are the most affected by lowering the precision, especially for HBFP4. Thus, we keep the first and last layers of CNNs in HBFP6 during HBFP4 training to increase its accuracy. However, the increase in precision HBFP6 provides for the first and last layers still does not achieve enough optimization to reach FP32 accuracy. In Figure 2, the red dashed curve shows this configuration, and the curve gets sharper and lower compared to HBFP4-only training. However, the generalization and optimization power of the model is still unbalanced, leading to convergence to another bad local minima.

We introduce Accuracy Boosters, an epoch-driven mixed-mantissa HBFP that uses HBFP6 only in the last epoch and converts $99.7\%$ of all arithmetic operations in training to HBFP4. We hypothesize that using HBFP6 for the last epoch is sufficient to boost the accuracy, while the rest of the epochs are trained using HBFP4. We leverage the insight that last epochs have more effect on the final model's accuracy [13], [14], [28], [36]. We claim that training with 4-bit mantissas helps the model generalize and reach a certain loss value. Afterward, switching to 6-bit mantissas helps the model optimize and fine-tune in the final epochs and increase accuracy to the FP32 level. The loss landscape for Accuracy Boosters (the red solid curve in Figure 2) supports our hypothesis. We see that the curve gets really close (note that the plot is in log scale, thus $-2$ is closer to $-4$ compared to $0$) to HBFP6 and FP32 curves and finally achieves FP32 accuracy.

## IV. Experimental Results

We experiment on the state-of-the-art models and datasets for various DNN tasks to test our hypotheses. We train ResNet20/74 [15], and DenseNet40 [16] on CIFAR10 and CIFAR100 [19] datasets for image classification. We also train a Transformer-Base [33] on the WMT16 English-German dataset for machine translation. Models trained on CIFAR10 are trained for 160 epochs, whereas for CIFAR100, the total number of epochs for all models is 300. The transformer is trained for 70 epochs. We use FP32 as the baseline for both model accuracies and hardware comparisons. For the image classification experiments, we report the Top1 validation accuracies; for machine translation, we report the BLEU scores. Moreover, to show the impact of our method, we tune the hyperparameters of FP32 models and then train the same models from scratch with the same hyperparameters in HBFP, showing that our method can be used out of the box without further hyperparameter tuning.

We use an analytic model to estimate the area of the most basic operation in DNN training—Dot product followed by activation unit for different encodings. Fixing the operation enables us to compare the arithmetic density ((operations/s)/area) solely on the amount of area. Thus, we define the gain in arithmetic density to be the same as the gain in the area. For FP32 dot product units of size $N$, we estimate the hardware cost as the sum of the cost of $N - 1$ FP32 adders, $N$ FP32 multipliers, one FP32 accumulator (adder), and one floating-point activation unit. For HBFP dot product units, we estimate the hardware cost as the sum of the cost of $N - 1$ fixed-point adders, $N$ fixed-point multipliers, one FP32 accumulator (adder), one floating-point activation unit, and one adder for signed exponents. We also add the costs of conversions between FP32 and fixed-point numbers by modeling the converter blocks.

### A. Minimizing HBFP

Table I shows the Top1 validation accuracies for ResNet20, ResNet74, and DenseNet40 on CIFAR10 and CIFAR100 datasets trained with various HBFP configurations. We observe that HBFP6 is the smallest HBFP configuration that gives accuracies within 2% of FP32 accuracy for block sizes up to 256. Larger blocks will contain a larger variety of values in terms of magnitude (affected e.g., by outliers), so it will result in larger approximation errors than smaller blocks and lower accuracy in training.

We also report HBFP4 accuracies to show the limitations of HBFP. Even for the small models like ResNet20, with a block size of 16, the accuracy drops more than 9%. As the accuracy drop for ResNet74 and DenseNet40 on CIFAR100 is considerably high even with HBFP5 (not shown here for compactness), we did not train these models with HBFP4. We observe that for HBFP4, the sensitivity to the block size increases for all the models because the distortions in the tensor distributions increase (see Section II).

TABLE I: Top-1 validation accuracies of various CNN models for various HBFP configurations

| Number Format | Block /Area | Models and Datasets | | | |
| --- | --- | --- | --- | --- | --- |
| | | CIFAR10 | | CIFAR100 | |
| | | ResNet20 | ResNet74 | ResNet74 | DenseNet40 |
| **FP32** | - | 91.72 | 93.57 | 74.55 | 72.42 |
| HBFP8 | 576/10.0 | 91.52 | 93.36 | 74.32 | 73.73 |
| HBFP6 | 16/11.2 | 91.12 | 93.38 | 73.51 | 72.08 |
| | 25/12.3 | 91.09 | 92.54 | 73.20 | 71.77 |
| | 36/13.1 | 91.29 | 92.61 | 72.87 | 71.83 |
| | 49/13.6 | 91.33 | 92.93 | 72.40 | 71.87 |
| | 64/13.9 | 91.12 | 92.93 | 72.40 | 71.81 |
| | 256/14.8 | 91.38 | 92.79 | 72.53 | 71.50 |
| | 576/15.0 | 90.65 | 92.19 | 72.51 | 71.02 |
| HBFP4 | 16/15.5 | 82.59 | 76.85 | - | 63.70 |
| | 25/17.8 | 81.82 | 78.62 | - | 64.25 |
| | 36/19.3 | 80.84 | 76.64 | - | 63.34 |
| | 49/20.4 | 79.32 | 71.19 | - | 65.55 |
| | 64/21.3 | 80.18 | 74.35 | - | 62.37 |
| | 256/23.4 | 76.96 | 60.65 | - | 60.02 |
| | 576/23.9 | 75.33 | 66.70 | - | 59.77 |
| **Total Number of FLOPs required to train the model** | | 41M | 174M | 326M | 542M |

### B. Accuracy Boosters

Considering HBFP hardware model, a block size of 64 is within 90% of the maximum area/power gain while achieving accuracies with less than 1% degradation for HBFP6. Thus, we choose block size of 64 as the sweet spot and test Accuracy Boosters using this block size. We perform the last epoch of the training in HBFP6 and the rest in HBFP4 for all the experimental settings. We also trained by keeping the last 10 epochs in HBFP6 to observe the improvement in accuracy for the CNN models. We keep all CNN models' first and last layers in HBFP6. The first and last layers of the CNN models account for a negligible amount of computation; thus, keeping them in slightly higher precision during HBFP training does not result in a significant increase in the hardware area or energy consumption. We can see that for most of the CNN models, Accuracy Boosters outperforms FP32. When we keep the last 10 epochs in HBFP6, we observe that the accuracies slightly increase (see Table II).

TABLE II: Top-1 validation accuracies of various CNN models for Accuracy Boosters

| Epochs using HBFP6 | Models and Datasets | | | |
| --- | --- | --- | --- | --- |
| | CIFAR10 | | CIFAR100 | |
| | ResNet20 | ResNet74 | ResNet74 | DenseNet40 |
| Only last | 91.24 | 92.62 | 73.74 | **73.61** |
| Last 10 | 91.36 | 93.02 | 74.28 | **74.10** |
| **FP32** | 91.72 | 93.57 | 74.55 | 72.42 |

Table III shows the results of applying Accuracy Boosters to

the Transformer. We observe that for the Transformer, HBFP6 performs similarly to FP32. While standalone HBFP4 does not incur a significant accuracy loss, Accuracy Boosters still help further bridge the gap to FP32 and even outperform it.

TABLE III: BLEU Scores for Transformer-Base trained on IWSLT'14 De→En task for various encodings

|  | FP32 | HBFP6 | HBFP4 | Booster |
|---|---|---|---|---|
| BLEU Score | 34.77 | 34.47 | 32.64 | 36.08 |

We observe that mixed-mantissa training using Accuracy Boosters can be carried out on arithmetic units designed for HBFP4. The small fraction of total training operations that use HBFP6 can be bit-sliced to fit on the 4-bit arithmetic units, similar to techniques proposed in prior work [38], while maintaining the same throughput. Thus, we claim the arithmetic density of a hardware accelerator using Accuracy Boosters will be approximately equal to the arithmetic density of HBFP4.

In conclusion, Accuracy Boosters offers up to $21.3\times$ higher arithmetic density compared to FP32 by using only 4 bits for $99.7\%$ of total training computations while achieving comparable or better accuracy. Our analytic model estimates another state-of-the-art reduced precision format—BFloat16 only offers $4.9\times$ higher arithmetic density compared to FP32. Hence, the much superior arithmetic density of HBFP4 enables Accuracy Boosters to offer a further $4.4\times$ higher arithmetic density compared to BFloat16 . Apart from arithmetic density, 4-bit mantissas promise significant memory savings, but the exact value depends on the layout and scheme and is outside the scope of this work.

## V. RELATED WORK

In recent years, there has been a significant amount of research on inference and training [4], [5], [8], [17], [21], [23], [29], [39] with narrow numerical representations. Google Brain's bfloat16 [35], NVIDIA's mixed-precision training with FP16 [25], and another mixed-precision scheme using FP8 [31] are the most commonly-used ones. Recent research advocates the use of Block Floating-Point for DNN training [11] and inference [6]. Flexpoint [20] and Dynamic Fixed-Point [7] propose block-floating-point formats for training with a 16-bit mantissa and a shared exponent. Prior work proposed a novel format for training DNNs with BFP, called Hybrid Block Floating-Point (HBFP) [10]. In this paper, we argue that reducing the mantissa bit width in HBFP significantly improves silicon efficiency while designing hardware for DNN training.

Many have proposed techniques to compensate for the data loss introduced by narrower numerical representations [12], [24], [31], [32]. Mixed-precision training has emerged as a popular technique to recover the information loss caused by quantization. Several techniques vary the precision layer-wise by using higher precision arithmetic for layers with greater significance [18], [30], [37]. Specifically, [3], [24], [34], [39]

use FP32 for the first and last layers. [13] employ fixed-point arithmetic with different bit widths epoch-wise over the course of training. Combining the layer-wise and epoch-wise approaches, [14], [27], [38] vary the precision adaptively per epoch and layer at the same time using control mechanisms. While all the aforementioned studies employ leaner arithmetic for a fraction of the training process, they fail to make leaner arithmetic the common case of the training process.

Recent work [9] suggests that during mixed-precision FP16 training, the optimizer states can be reduced to 8 bits by using a block-wise quantization method. This observation is in line with our work that applies quantization by extracting the largest exponent per block. Similarly, FAST [38] uses a block-floating-point-based layer-wise mixed precision approach using 2 and 4-bit mantissa. Unlike our work, FAST requires fine-tuning several additional hyperparameters for its training algorithm, making it difficult to apply to other DNN models. Another block-floating-point-based work, FlexBlock [27], uses 4 and 8-bit mantissa with various block sizes and also needs higher-precision block-floating-point formats only for weight gradient calculations that suffer more from quantization errors.

## VI. CONCLUSION

Several low-precision training techniques and specialized numerical formats have been introduced over the past decade to increase the arithmetic density of the DNN accelerators. One such format, Hybrid Block Floating-Point (HBFP), which allows for a majority of the DNN's arithmetic operations (i.e., dot products) to be performed using fixed-point arithmetic has been shown to achieve FP32 accuracy with 8-bit mantissa. However, a smaller number of mantissa bits allow for exceptional improvements in arithmetic density. In this paper, we perform a full-scale exploration of the HBFP design space for emerging models and datasets. We show that HBFP6 is the smallest HBFP format achieving FP32 accuracy for all block sizes. We propose the Accuracy Boosters technique to bring HBFP4 into training, using HBFP6 in the last epoch, leveraging the insight that each epoch has a different effect on training. We show that the last stage of training requires more precision than the rest. We use an analytic model to show that our method achieves up to $21.3\times$ higher arithmetic density over FP32 and $4.4\times$ higher density over BFloat16 , while maintaining or outperforming FP32 accuracy.

## ACKNOWLEDGEMENTS

The authors thank the anonymous reviewers and the members of PARSA at EPFL for their precious comments and feedback. We would also like to thank Nicholas Sperry for his contributions to the loss landscape experiments. This work has been partially funded by a Microsoft PhD Fellowship, and the following grant: "Unified Accelerators for Post-Moore Machine Learning" from the Swiss National Science Foundation (SNSF).

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
