# OpenReview forum: "Accuracy Boosters: Epoch-Driven Mixed-Mantissa Block Floating Point for DNN Training"
_iscaconf.org/ISCA/2023/Workshop/ASSYST — ASSYST Oral_

### Official Review · Reviewer_YVLu · 2023-05-04
**A good paper with promising idea. Some experiments results on STOA models would make the paper stronger.**

**Rating:** 6
**Confidence:** 4

**Review:**

**Summary**

The author propose a new way to train mixed-precision model. The results shows power saving while keeping comparable accuracy with FP32.

**Review (Strengths/Weaknesses):**

**Strengths**
* The related works are well referenced. The authors are knowledgable at this domain.
* The proposed method is novel and has many potential to be further extended in follow up works.
* The paper is well written. It is structured and easy to follow.

**Weaknesses**
* My main concern is the experiments:
1) I would want to see the performance on larger dataset. CIFAR10 and CIFAR100 are not sufficient. Not using a larger dataset would make reader suspicious about is there any restriction of the methods, e.g., scalability.
2) I would expect some comparisons with related works in the experiments, e.g., FlexBlock, ref-10, ref-11. I understand that the authors has set up the baseline of FP32, but it is hard to judge the improvement over related works when only using FP32 as baseline.
3) It is true that using less-bit to train the model can infer consuming less energy. However, I don't know if I can trust the number reported in Table IV. How the power consumption is calculated need to be stated more clearly. Especially, when the energy saving is the main highlight of this work.
* I would appreciate if the authors can present some figures/flowcharts/algorithms about their proposed method.

**Reviewer Expertise:**

Knowledgeable: I used to work in this area and/or I try to keep up with the literature but might not know the latest developments.

---

### Official Review · Reviewer_y9MC · 2023-05-06
**Interesting paper, but writing and evaluation could be stronger**

**Rating:** 6
**Confidence:** 3

**Review:**

This paper discusses interesting insights regarding the high precision requirement (spatial: first and last layers, temporal: last epoch) and the loss space of various precision of HBFP. Based on that, this paper proposes a mixed-precision HBFP approach for training.

Although the insights and ideas are interesting, this version of the paper needs more technical clarifications and evaluation data to better support the ideas of this paper. Please see the following for details.

1. Which distance metric (e.g., L1 norm, L2 norm, etc.) did this paper use in the Wasserstein distance formulation? Can you clarify why did this paper use that particular distance metric?
2. Some data to support why Wasserstein distance-based approach is better than KL divergence-based one will be helpful.
3. Please explicitly discuss that A and B in "HBFP(A,B)" (x-axis labels of Fig 1) mean the # mantissa bits and block size.
4. Can you clarify which loss function Fig. 2 plotted?
5. [Minor] There are duplicated labels ("Booster (last)") in Table III; the last one seems to be "Booster (last 5)." For clarity, "Booster (last 1)" and "Booster (last 5)" can be better labels.
6. ImageNet data on the CNN side would be interesting to see the impact on a more complex problem.
7. Can you clarify the evaluation setting of power results? Is it from a CPU, GPU, or an accelerator?
8. As a part of the motivation was in the area and energy efficiency, those data are desired. (Regarding power results: power is not directly translated into energy)
9. More evaluation platforms for area and energy data are desired (e.g., CPU, GPU, and accelerator)
10. More evaluation on the Transformer side is desired considering the topic of this workshop; currently, the evaluation includes only one model.

**Review (Strengths/Weaknesses):**

# Strength
+ Interesting insights and ideas
+ Promising results

# Weakness
- Many missing pieces of information
- Limited evaluation
- Weak focus on the workshop's topic (architecture and system support for Transformer models)

**Reviewer Expertise:**

Knowledgeable: I used to work in this area and/or I try to keep up with the literature but might not know the latest developments.

---

### Official Review · Reviewer_eApU · 2023-05-06
**Adequate study on minimizing HBFP, but lacks novelty and requires improvement in writing quality**

**Rating:** 4
**Confidence:** 4

**Review:**

The paper under review investigates the reduction of bit numbers in Hybrid Block Floating-Point (HBFP) through experimental study. The authors explore a broad parameter space and utilize varying precision for different layers and epochs to achieve this reduction. The conversion of 99.7% of arithmetic operations to 4-bit greatly reduces power consumption.

Pros:
1. The Accuracy Booster design is well-constructed, and the experimental results are impressive in reducing the bit numbers of Hybrid Block Floating-Point (HBFP).
2. The use of Wasserstein distance and loss landscapes is a valuable approach to demonstrate the motivation for the accuracy booster design.

Cons:
1. The paper's contributions lack novelty, as the use of different precision for different layers and epochs has been extensively explored in prior research.
2. The background information on HBFP in Section II is insufficient, which may impede readers' understanding of the paper.
3. The experimental setup is not well-described, particularly with regards to the methodology for measuring power consumption.

In summary, the writing quality could be improved, and the paper's novelty is insufficient. Therefore, based on the identified cons, the paper is not recommended for acceptance.

**Review (Strengths/Weaknesses):**

Strengths:
1. The Accuracy Booster design is well-constructed, and the experimental results are impressive in reducing the bit numbers of Hybrid Block Floating-Point (HBFP).
2. The use of Wasserstein distance and loss landscapes is a valuable approach to demonstrate the motivation for the accuracy booster design.

Weakness:
1. The paper's contributions lack novelty, as the use of different precision for different layers and epochs has been extensively explored in prior research.
2. The background information on HBFP in Section II is insufficient, which may impede readers' understanding of the paper.
3. The experimental setup is not well-described, particularly with regards to the methodology for measuring power consumption.

**Reviewer Expertise:**

Knowledgeable: I used to work in this area and/or I try to keep up with the literature but might not know the latest developments.